# Exploring the Gut Microbiota–Muscle Axis in Duchenne Muscular Dystrophy

**DOI:** 10.3390/ijms25115589

**Published:** 2024-05-21

**Authors:** Debora Mostosi, Monica Molinaro, Sabrina Saccone, Yvan Torrente, Chiara Villa, Andrea Farini

**Affiliations:** 1Stem Cell Laboratory, Dino Ferrari Center, Department of Pathophysiology and Transplantation, University of Milan, 20122 Milan, Italy; debora.mostosi@unimi.it (D.M.); yvan.torrente@unimi.it (Y.T.); chiara.villa2@unimi.it (C.V.); 2Neurology Unit, Fondazione IRCCS Ca’ Granda Ospedale Maggiore Policlinico, 20122 Milan, Italy; monica.molinaro@outlook.it (M.M.); sabrina.saccone21@gmail.com (S.S.)

**Keywords:** gut microbiota, muscular inflammation, Duchenne muscular dystrophy, muscle wasting

## Abstract

The gut microbiota plays a pivotal role in maintaining the dynamic balance of intestinal epithelial and immune cells, crucial for overall organ homeostasis. Dysfunctions in these intricate relationships can lead to inflammation and contribute to the pathogenesis of various diseases. Recent findings uncovered the existence of a gut–muscle axis, revealing how alterations in the gut microbiota can disrupt regulatory mechanisms in muscular and adipose tissues, triggering immune-mediated inflammation. In the context of Duchenne muscular dystrophy (DMD), alterations in intestinal permeability stand as a potential origin of molecules that could trigger muscle degeneration via various pathways. Metabolites produced by gut bacteria, or fragments of bacteria themselves, may have the ability to migrate from the gut into the bloodstream and ultimately infiltrate distant muscle tissues, exacerbating localized pathologies. These insights highlight alternative pathological pathways in DMD beyond the musculoskeletal system, paving the way for nutraceutical supplementation as a potential adjuvant therapy. Understanding the complex interplay between the gut microbiota, immune system, and muscular health offers new perspectives for therapeutic interventions beyond conventional approaches to efficiently counteract the multifaceted nature of DMD.

## 1. The Bidirectional Gut–Muscle Axis

Comprising 10^14^ microbial cells within the digestive tract, the gut microbiota is categorized into various species, families, and phyla. This intricate community comprehends not only bacteria, but also eukaryotes and viruses, that establish a synergic communication among themselves and with the host, largely influencing human physiology, homeostasis, and health. Albeit a far more in-depth knowledge about bacterial components is reported in the literature, recent studies have focused on the eukaryotic communities and the consortium of viruses, forming the so-called human virome harbored in the digestive system [1].

Due to this cardinal role in human health and disease, the gut microbiota is sometimes named as our “forgotten organ”. The impact of the gut microbiota on human well-being is partially attributable to its co-evolvement with the host to reciprocally satisfy biological and biochemical needs [2]. Indeed, it is implicated in numerous metabolic processes, including energy production and storage, as well as the fermentation and absorption of undigested carbohydrates. These functions probably led to a vigorous evolutionary driving force toward the development of a symbiotic relationship between humans and gut bacteria [1]. Fundamental for host homeostasis, the interactions between the gut microbiota and the host extend beyond the digestive system, reaching organs such as the cardiovascular system, brain, skin, pancreas, and skeletal muscles [2,3]. Many studies focused on the effects of the gut microbiota on the modulation of the diverse components of the immune system, liver, and intestinal metabolism, but also on behavior; nevertheless, few studies investigated the influence of the gut microbiota on skeletal muscle, the largest metabolic organ, representing approximately half of the total human body weight [2]. The microbiota significantly influences skeletal muscle functions and health, regulating processes impacting on pathophysiology—including oxidative stress, mitochondrial function, neuromuscular connectivity, and insulin resistance— inflammation, and immunity, specifically the activation of systemic and chronic inflammatory cues and of signaling pathways mainly depending on the expression of toll-like receptor (TLR)-4, nuclear factor-κB (NF-kB), and c-Jun N-terminal kinase phosphorylation [4]. Recent research suggested a complex bidirectional crosstalk between intestinal flora and skeletal muscles, although the underlying regulatory pathways remain unclear. The endocrine properties of skeletal muscles determine the release of myokines, cytokines, and proteins, exerting systemic effects on various body systems, including the digestive system and the gut microbiota [5]. For instance, irisin was recently identified as a myokine mostly produced and secreted into the circulation by adipose and skeletal muscle tissue [6]; the knockout of fibronectin type III domain-containing 5 (Fndc5)/irisin in mice caused altered diversity and richness of gut microbiota [7]. In return, gut bacteria influence skeletal muscle homeostasis through a multitude of signaling pathways, establishing a bidirectional gut–muscle axis. Studies using murine animal models provided compelling evidence supporting this finding, demonstrating skeletal muscle atrophy and reduced mass in germ-free mice lacking gut microbiota, while the transplantation of microbiota from pathogen-free mice alleviates muscle atrophy [2]. Gut microbiota depletion in mice through antibiotic treatment results in muscle atrophy and decreased endurance [8]; additionally, Ghrelin-deficient mice with pro-inflammatory gut microbiota exhibit reduced muscle mass and function as they age [9].

While the current literature primarily focuses on animal models, altered gut microbial status is evident in elderly individuals, sarcopenic, and cachectic patients. In patients affected by liver pathology, altered *Firmicutes/Bacteroidetes* ratios and increased Gram-negative bacteria correlate with a lower muscle mass [10]. A longitudinal study correlates age-related frailty with microbiota richness, affecting essential amino acid, nitrogenous base, and vitamin B production [11]. Reductions in *Lactobacilli, Bacteroides*, and *Prevotella*, coupled with an increase in *Enterobacteriaceae*, are associated with frailty symptoms in the elderly population [12]. Athletes exhibit distinct gut microbiota due to protein intake differences [13] and consumption habits [14], revealing the potential role of the microbiota as a human health biomarker [5]. The interindividual variability in gut microbiota composition can affect muscle mass and function: increased numbers of *Oscillospira* and *Ruminococcus* microbial taxa and reduced numbers of *Barnesiellaceae* and *Christensenellaceae* are observed in older adults with physical frailty and sarcopenia [15]. Furthermore, in vitro studies revealed that muscle mass can be directly influenced by gut microbial products: for instance, indoxyl sulfate is inversely correlated with muscle strength, since it stimulates muscle atrophy by promoting inflammation, oxidative stress, and myasthenic gene expression [4].

Autophagy has recently gained recognition as a crucial process in regulating skeletal muscle function by maintaining homeostasis through the degradation of altered or damaged organelles. The involvement of the AMP-activated protein kinase (AMPK) and peroxisome proliferator-activated receptor coactivator-1 (PGC-1) signaling pathways in autophagy and inflammation underscores their association with the gut microbiota–muscle axis. As age advances, the activation of AMPK and PGC-1 declines. The inhibition of these signaling pathways not only diminishes autophagic activity but also amplifies the inflammatory response, contributing to impaired skeletal muscle function. The reduction in autophagic activity leads to the accumulation of dysfunctional organelles in senescent cells, resulting in an increase in reactive oxygen species (ROS) production. This elevated ROS levels, in turn, promote the activation of the Nod-like receptor 3 (NLRP3) inflammasome: the NF-κB signaling pathway stimulates the production of NLRP3, forming a complex interplay between autophagy, inflammation, and the loss of skeletal muscle function [4].

Dysbiosis can lead to the disruption of the intestinal epithelial barrier (IEB) [16]. The intestinal flora actively regulates IEB permeability and integrity through various mechanisms, such as the enhancement in arachidonoylglycerol (2-AG) and oleoylglycerol (2-OG) levels by short-chain fatty acids (SCFAs), thereby improving IEB integrity. However, dysbiosis damages epithelial cells and intercellular junctions in the IEB [17], facilitating bacterial translocation and the secretion of mucin with the activation of the lymphoid tissue associated with the intestinal layer, thereby triggering the recruitment of naive T and B cells and initiating the activation of innate defense mechanisms. These events trigger an inflammatory cascade that sustains the development of chronic inflammation [18]. Altered bowel permeability also facilitates the passage of microbial products like indoxyl sulfate (IS) and lipopolysaccharide (LPS) into the circulation, inducing systemic inflammation and impairing muscle function [4]. Dysbiosis further impairs SCFA synthesis. IS, a bacterial metabolite of tryptophan, exhibits pro-inflammatory effects in chronic kidney disease (CKD) patients: it increases inflammatory cytokines (TNF-α, IL-6, and TGF-β1) and ROS levels in C2C12 cells, accelerating skeletal muscle atrophy and negatively affecting muscle strength [16]. The chronic administration of IS significantly reduces skeletal muscle mass in mice [18].

LPS, an endotoxin in the outer membrane of Gram-negative bacteria, possesses pro-inflammatory properties. Altered IEB permeability may lead to LPS translocation into the circulatory system [19]. LPS binds to TLR4, causing the activation of the pro-inflammatory NF-κB pathway that—in turn—promotes the degradation of muscle proteins, induces inflammation and fibrosis, and impairs muscle regeneration after injury/atrophy [20]. TLR4 signaling also increases immune cells’ expression of TNF-α and IL-6, inducing systemic inflammatory responses [20]. Higher plasma levels of TNF-α and IL-6 are correlated with a reduction in muscle mass and strength in well-functioning older men and women [19]. Similarly, these pro-inflammatory cytokines are elevated in cancer cachectic patients and in rodent models of cancer cachexia. The chronic, low-level elevation of circulating IL-6 induces muscle atrophy, promoting a down-regulation of growth factor-mediated intracellular signaling that favors a more catabolic profile in muscles [20]. The microbiota-dependent inhibition of the production of TNF-α and IL-6 is associated with an improvement in muscle atrophy [16]. During prolonged intense exercise, circulating LPS stimulates significant ROS production, inducing increased oxidative stress that can cause insulin resistance, mitochondrial dysfunction, apoptosis, or autophagy in skeletal muscles [20]. LPS has been shown to significantly reduce the generation of multi-nucleated myotubes and inhibit myogenic differentiation in vitro [18].

Over the past decade, recognizing the presence of intestinal microbes in tissues beyond the gut has prompted investigation into the phenomenon of microbial translocation, initially observed in animal models. This led to the assessment of colorectal cancer patients for the presence of various microbial fragments, directly correlating with survival rates and disease progression [21]. Similarly, these phenomena were described in a plethora of inflammatory pathologies, such as inflammatory bowel disease (IBD), fatty liver disease, and pancreatitis [22].

These findings suggest that a disrupted IEB may allow the leakage of bacterial components or even bacteria into the circulatory system, exerting detrimental effects on distal tissues, including the skeletal muscle [18].

## 2. Interplay of Factors in Regulating Skeletal Muscle Physiology and Immune Balance

### 2.1. The Peroxisome Proliferator-Activated Receptors (PPARs)

The peroxisome proliferator-activated receptors (PPARs) belong to the family of nuclear receptors, which are ligand-regulated transcription factors activated by specific natural ligands, such as steroid hormones (e.g., estrogen and progesterone), lipids, retinoic acid, oxysterols, and thyroid hormones [23]. The PPAR subfamily comprises three nuclear receptor isoforms, PPARα, PPARβ/δ, and PPARγ, encoded by three distinct genes, situated on independent chromosomes in vertebrates [24]. PPARα is expressed in the heart and skeletal muscle, but also in the liver, kidney, and brown adipose tissues [24]: it regulates cellular uptake, the β-oxidation of fatty acids and energy homeostasis, and modulates glucose metabolism and inflammation [25]. PPARβ/δ is ubiquitously expressed, but it is particularly present in metabolically active tissue, primarily in organs and cells involved in fatty acid metabolism, as the skeletal muscle and myocardium [26]. It is expressed also in muscle satellite cells (SCs), and it is the prevalent PPAR isoform in skeletal muscle tissue [24]. PPARβ/δ regulates energy expenditure, tissue repair and regeneration, inflammation, and the myofiber type switch associated with physical exercise [26]. PPARγ is associated with genes that affect cellular energy homeostasis and insulin action. It is necessary and sufficient to induce adipocyte differentiation and controls the expression of the genes involved in lipid uptake by adipocytes, playing a fundamental role in adipogenesis and triglyceride storage [25]. It is therefore involved in the deposition of fat in several organs, including the skeletal muscle. PPARs play a key role in muscle pathophysiology, being involved in cancer cachexia, aging, myopathies, and muscular dystrophies, as well as respiratory and cardiovascular diseases [24]. The pharmacological activation of PPARβ/δ using GW501516, a PPARβ/δ agonist, increases utrophin A expression in a murine model of DMD—the mdx mice—by binding to the PPRE in the promoter region of utrophin A. GW501516 treatment improved mdx sarcolemmal integrity and protected mdx skeletal muscles against eccentric contraction (ECC)-induced damage. Consequently, the pharmacological activation of PPARβ/δ could potentially inspire new therapeutic strategies for DMD patients’ treatment [27].

PPARα, PPARβ/δ, and PPARγ are expressed along the gastrointestinal tract and can interact with the gut microbiota at various levels. For example, in the intestines, specific gut microbial species can regulate the expression of PPARγ, which in turn can change the composition of the gut microbiota [24]. The interactions of gut microbiota and PPARs also impact skeletal muscle physiology and pathology. A recent study showed that mice treated with metronidazole displayed gut dysbiosis, the overexpression of PPARγ and its target gene adiponectin in skeletal muscle, and skeletal muscle atrophy [28]. In conclusion, causal connections between PPARs, gut microbiota, and muscle pathophysiology are progressively being discovered, but further studies to clarify these interactions are needed. The creation of tissue-specific PPAR KO models will allow the evaluation of the role of these receptors in muscle homeostasis [24].

### 2.2. Gut-Derived Metabolites

In addition to the extraordinary abilities of the gut microbiota to synthesize a plethora of enzymes, a wide range of metabolites are produced by host microorganisms such as those derived from diets—short-chain fatty acids (SCFAs), choline derivatives, polyamines, tryptophan; those generated by the host and modified by the gut microbiota; those synthetized de novo by intestinal microbes, such as polysaccharide A [29]. Intriguingly, these microbiota-derived mediators regulate the functions of the majority of the tissues, including the skeletal muscle.

### 2.3. Short-Chain Fatty Acids (SCFAs)

Given the importance of gut microbiota homeostasis for skeletal muscle physiology, an imbalance in its composition, known as dysbiosis, can contribute to skeletal muscle wasting through multiple pathways, including inflammatory processes. Animal models of acute inflammation exhibit reduced protein synthesis (via the mTOR-regulated initiation of translation), increased muscle proteolysis (ubiquitin-proteasome pathway), the stimulation of cell apoptosis, and the inhibition of SC differentiation [20]. Persistent inflammation is recognized as a key contributor to skeletal muscle function loss [16], impairing SC function and muscle regeneration [30]. The intestinal flora influences immune cell maturation and function, affecting CD4+ and CD8+ T-cell development and differentiation. Substances produced by the gut microbiota, such as SCFAs, indirectly modulate immune function [17]: in line with this, acetate, propionate, and butyrate exhibit anti-inflammatory effects by binding to G-protein-coupled receptor 43 (GPR43) in immune cells, limiting histone deacetylase (HDAC) activity, and modulating cytokine and prostaglandin E2 (PGE2) synthesis [31].

Butyric acid antagonizes colonic inflammation by inhibiting the NF-κB activation and activating PPARγ, reducing colonic permeability and inflammation [32]. Comparisons between elderly patients with and without primary sarcopenia revealed a significant depletion of SCFA-producing bacteria in sarcopenic subjects [33]. SCFAs also regulate the lipid, carbohydrate, and protein metabolisms in skeletal muscles, affecting fatty acid uptake and oxidation, skeletal muscle lipid metabolism, and promoting glucose uptake and glycogen synthesis [18,34]. Enhancing the production of SCFAs by gut bacteria has the potential to positively impact skeletal muscle mass in humans [35].

The indirect effects of SCFAs on muscles include increased blood flow and anti-inflammatory properties via epigenetic mechanisms, acting through the phosphoinositide 3-kinase (PI3K)/protein kinase B (AKT)/mTOR/GLUT4/forkhead box O (FOXO) signaling cascade or by blocking HDACs and regulating NF-κB-dependent genes [20].

The synergy between the direct activation of innate and adaptive immunity by the intestinal flora and the indirect modulation of the immune response by microbiota-derived metabolites maintains intestinal immune homeostasis [17]. Therefore, dysbiosis characterized by microbial imbalance and reduced diversity can lead to chronic inflammation through various mechanisms.

### 2.4. Choline Derivatives

Choline, a member of the vitamin B group, plays a pivotal role in the liver and neural metabolisms. It facilitates the synthesis and metabolism of essential membrane constituents, such as phospholipids and triglycerides. Studies have shown that choline deficiency significantly impacts the integrity of skeletal muscle cells, resulting in the up-regulation of creatine phosphokinase secretion [36]. In addition, it was proposed a role for the phosphatidylcholine in the production of a choline precursor and the subsequent synthesis of the neurotransmitter acetylcholine (ACh) [37], the regulator of skeletal muscle contraction [38].

### 2.5. Polyamines

Polyamines are produced through the decarboxylation of amino acids, playing diverse roles in cell growth, metabolism, and development, including post-translational modifications [39]. They exhibit antioxidant, anti-inflammatory, and anti-apoptotic properties, influencing autophagy and mitochondrial apoptosis. Notably, polyamines are down-regulated in aged animals [40] and play a crucial role in regulating muscle atrophy and muscle fiber size through the rapamycin complex 1 (mTORC1), as well as the expression of spermine and spermidine synthase [41].

### 2.6. Tryptophan (Trp)

Tryptophan (Trp) metabolism is closely associated with frailty and sarcopenia and is influenced by the gut microbiota. The intestinal flora metabolizes unabsorbed Trp and plays a role in regulating Trp metabolism pathways, such as kynurenine (Kyn), 5-hydroxytryptamine (5-HT), and indole, within the gastrointestinal tract, thereby controlling the production of their metabolites. Metabolites of Trp derived from the gut microbiota can exacerbate the development of age-related frailty and sarcopenia by triggering inflammation in the gastrointestinal tract, nervous system, and muscles. In older adults experiencing physical frailty and sarcopenia, serum Trp levels have been identified as distinguishing factors from the control group. Additionally, the relative concentration of indole in the serum tends to be lower in frail older adults compared to non-frail individuals, with changes in indole levels associated with bacterial dysbiosis. Serum metabolomic studies in sarcopenia patients revealed a reduction in Trp levels, while other evidence suggests a negative correlation between dietary Trp intake and the risk of sarcopenia [42]. Data from patients affected by diffuse large B-cell lymphoma proved that control mice fed a Trp-deficient diet exhibit signs of muscular atrophy, including reduced weight, the down-regulation of muscle fiber area, and an increase in serum myostatin expression [43].

### 2.7. Bile Acids

The primary bile acids (BAs) are synthetized in the liver from cholesterol, subsequently conjugated to glycine (predominantly in humans) or taurine (rodents), and after a meal, secreted into the duodenum. Here, BAs facilitate the digestion and absorption of dietary lipids and fat-soluble vitamins. In the intestine, bacteria carrying bile salt hydrolase deconjugate the conjugated primary BAs, which are then metabolized into secondary BAs by the gut microbiota [44]. The primary BA metabolism is therefore regulated by the gut bacteria. As a result, a BA dysmetabolism has been reported in germ-free (GF) animals [45]. Ultimately, microbiota-derived bile acids have been recognized as relevant cell-signaling molecules that can communicate with skeletal muscle tissue [46].

Microbially metabolized BAs can be released in the systemic circulation, acting as endocrine molecules that regulate the lipid and glucose metabolisms, energy expenditure, and systemic inflammation [46]. However, most BAs are reabsorbed in the distal ileum, with a consequent temporary increase in BA concentration in blood, and are subsequently transported to the liver, where they are recycled. This cycle is known as enterohepatic circulation [47]. BA signaling activity is mainly mediated by binding to specific cellular receptors, such as the nuclear farnesoid X receptor (FXR) and the Takeda G protein-coupled receptor 5 (TGR5, also known as G protein-coupled bile acid receptor 1, GPBAR1) [46].

Previous studies suggested that BAs, FXR and BAs/FXR-induced FGF19, may impact skeletal muscle mass and function [44]. Consequently, the exploration of a potential gut microbiota–bile acids–skeletal muscle axis has garnered increased attention [46]. Kobayashi et al. demonstrated that circulating BAs were associated with skeletal muscle volume in patients with non-alcoholic fatty liver disease (NAFLD) [48]. Furthermore, enterokine FGF19 improves skeletal muscle mass and strength through an increase in the myofiber size and ameliorates the skeletal muscle atrophy caused by aging, obesity, or glucocorticoids in mouse models [49]. In C2C12 mouse myoblasts, FGF19 can alleviate palmitic acid (PA)-induced mitochondrial dysfunction and oxidative stress through the AMPK/PGC-1α signaling pathway, factors that can impair the quality and function of the skeletal muscle [50]. Moreover, FGF19 improved muscle atrophy, impaired the lipid and glucose metabolisms, and decreased irisin levels in PA-treated myotubes and in the skeletal muscle of mice fed a high-fat diet [6]. In mouse models, the depletion of gut microbiota was associated with an aberrant BA metabolism in the intestine, a consequent inhibition of FXR-FGF15 signaling, and skeletal muscle atrophy. The treatment of Abx mice with FGF19 partly reversed skeletal muscle loss. These findings reveal that the depletion of the gut microbiota induces skeletal muscle atrophy through the BA-FXR-FGF15/19 axis [44]. In aged mice, the FXR agonist fexaramine improved skeletal muscle mass and performance, indicating that FXR-FGF15/19 signaling is a potential therapeutic target for sarcopenia [51]. Furthermore, TGR5 is expressed in skeletal muscles and, when TGR5 is activated, muscle cell differentiation and muscle hypertrophy are promoted. Accordingly, TGR5^KO^ mice showed reduced skeletal muscle mass and strength and a lower expression of differentiation and hypertrophy-related genes in the skeletal muscle. Increased muscle strength was reported in muscle-specific TGR5 transgenic mice [47].

### 2.8. Taurine

Taurine (2-aminoethanesulfonic acid) has a fundamental function in regulating the osmotic pressure of different tissues. According to its recognized functions in limiting oxidative stress, Taurine exerts cytoprotective and anti-aging activities, as demonstrated in TauTKO mice that presented with morphological defects in the skeletal muscle (as myofiber necrosis) and in protein folding, a reduced activity of the mitochondrial complex 1 complex, decreased muscle force, and the up-regulation of *Cyclin-dependent kinase 4 inhibitor A (p16INK4a)* expression [52].

All these compounds and their effects on the skeletal muscle are summarized in Table 1.

## 3. Therapies Targeting the Dysbiotic Microbiota

Given the contribution of gut microbiota alterations to muscle wasting, restoring the intestinal flora offers novel opportunities for treating various myopathies. Approaches to address gut microbiota imbalances include therapeutic fecal bacteria transplantation and the use of pro-/prebiotic products [53]. In a study, fecal microbiota transplantation from young rats to aged recipients alleviated age-related sarcopenia, preserving the gut barrier integrity and enhancing muscle mitochondrial function [54].

Chen et al. identified a reduction in muscle function and fiber size in a 5-fluorouracil (5-Fu)-induced malnutrition rat model with a distinct gut microbiota profile. Fecal microbiota transplantation from healthy rats to 5-Fu-treated rats inhibited muscle inflammation and improved muscle function [55]. Bindels et al. reported that a leukemic and cachexic mouse model showed gut microbiota depletion. Normalizing the gut microbiota through oral probiotic supplementation reduced inflammation, autophagy, and proteolysis markers in muscles, preventing skeletal muscle inflammation and atrophy [56]. Another study demonstrated that the dietary supplementation of obacunone in obese KKAy mice promoted muscle hypertrophy and prevented obesity and hyperglycemia [57].

In human triathletes, *Lactobacillus plantarum* PS128 supplementation limited the development of inflammatory cues; improved muscular performance by modulating gut microbiota composition and increasing the SCFAs’ acetate, propionate, and butyrate levels; and diminished the extent of oxidative stress and levels of creatine kinase, thioredoxin, and myeloperoxidase [58]. *Lactobacillus casei LC122* or *Bifidobacterium longum BL986* enhanced muscle strength and function, reduced oxidative stress and inflammation in peripheral tissues, and improved intestinal barrier integrity [59]. Therefore, fecal microbiota transplantation from healthy subjects or the modulation of the gut microbiota using prebiotics, probiotics, or synthetic bacteria could be considered as potential treatments to enhance muscle function [16].

## 4. The Pathogenesis of Duchenne Muscular Dystrophy

Muscular dystrophies encompass a group of genetic disorders characterized by defects in muscle proteins, leading to progressive weakness and muscle mass loss [5]. Duchenne muscular dystrophy (DMD) is the most common of the human muscular dystrophies, with a reported global incidence of 1 in 5000 live male births [60]. DMD is an X-linked recessive disease caused by mutations in the dystrophin gene, disrupting protein production. Mutations in DMD can also cause Becker muscular dystrophy (BMD), a milder disease characterized by the preserved production of a partially functional dystrophin [61]. In DMD, the respiratory muscles, such as the diaphragm, are typically affected earlier and more severely, causing sleep-disordered breathing, hypoventilation, and weakness of expiratory/inspiratory musculature [5], so that patients commonly succumb to either respiratory or cardiac failure, often in their second or third decade of life [61].

Dystrophin is part of the dystrophin-associated protein complex (DAPC) that connects the muscle cell cytoskeleton with the sarcolemma and the ECM, stabilizing muscle fibers. In the skeletal muscle, the N-terminal and C-terminal dystrophin domains connect cytoskeletal F-actin with the extracellular matrix, stabilizing the sarcolemma and protecting the myocytes from contraction-induced damage and necrosis. In DMD, dystrophin absence determines sarcolemma fragility—with sequential muscle wasting due to mechanical stress accumulated during the contraction/relaxation cycles—and affects the regenerative potential of SCs [5]. Dystrophin deficiency in SCs causes altered cell polarity, epigenetic regulation, and asymmetric division, associated with aberrant myogenic differentiation. This asymmetric division is further exacerbated by mitotic defects, including centrosome amplification, spindle orientation mistakes, and prolonged cell cycle. The repeated cycles of myofiber degeneration–regeneration with the consequent exhaustion of the SC pool lead to progressive skeletal muscle degeneration, accompanied by a substitution of striated muscle tissue with connective and adipose tissue [5], contributing to SC exhaustion in the DMD context [62]. Rather than the skeletal muscle, DMD is accompanied by mitochondrial dysfunctions, altered calcium homeostasis, impaired autophagy and apoptosis, oxidative stress, and stress and dysfunctions of the sarcoplasmatic reticulum (SR) and of immunological and inflammatory processes [63]. We describe some of these secondary pathophysiological processes that can be triggered by dysbiosis in DMD.

## 5. Ions’ Homeostasis

In addition to their role in regulating digestion and nutrient absorption, intestinal microorganisms play a crucial role in maintaining the proper concentration of ions in the intestinal fluid, essential for the composition and function of the gut microbiota. Moreover, numerous studies have highlighted the significant role of disrupted ion homeostasis in DMD. Instead of solely ensuring the structural integrity of the sarcolemma, dystrophin and other members of the DAPC are involved in regulating various signaling molecules, including ion channels and their regulators. The evidence suggests that the DAPC plays a pivotal role in maintaining calcium balance and controlling the activity of sodium and potassium transport channels. Specifically, the DAPC interacts with the sarcolemmal calcium channels TRPC1 and TRPC4, the sodium channels Nav1.4 and Nav1.5, and the potassium channels Kir2 and Kir4.1. In DMD, the absence of dystrophin and the altered DAPC functionality leads to the impaired function of these ion channels, contributing to muscle damage [64].

### 5.1. Calcium

In DMD cells and muscles, the resting cytosolic Ca^2+^ concentration is higher than the Ca^2+^ level in physiological conditions [65]. This calcium overload can be ascribed to various causes, including altered Ca^2+^ influx from the ECM to the cytosol and an impairment in Ca^2+^ cycling between the SR and the cytosol [66]. Maintaining Ca^2+^ homeostasis is crucial for efficient muscle contraction and relaxation cycles. In both skeletal and cardiac muscle, the action potential triggers the activation of sarcolemmal L-type Ca^2+^ channels (CaV1.1 and CaV1.2 in the skeletal and cardiac muscles, respectively). In the cardiac muscle, the resulting Ca^2+^ influx activates the ryanodine receptor isoform RyR2 on the SR membrane, leading to the release of Ca^2+^ from the SR. In the skeletal muscle, CaV1.1 and RyR1 physically interact, promoting the opening of RyR1 upon CaV1.1 activation [66]. Skeletal muscle contraction is also regulated by the store-operated Ca^2+^ entry (SOCE) machinery, located in the triad of the skeletal muscle and constituted by various proteins, including stromal interaction molecule 1 (STIM1), Orai, transient receptor potential canonical (TRPC) channel, and RyRs. In both the myocardium and skeletal muscles, the released Ca^2+^ binds to troponin C, triggering muscle contraction, and then it is re-sequestered into the SR lumen by sarco-endoplasmic reticulum calcium ATPase (SERCA) proteins [66].

Dystrophin absence and DAPC impairment weaken the plasma membrane of muscle cells, making it more susceptible to microtears during mechanical stress. Consequently, the permeability of the sarcolemma for ions increases. The appearance of microtears could determine increased Ca^2+^ intake into muscle cells [64], but other mechanisms have been described as the most relevant in determining excessive Ca^2+^ entry in DMD. Voltage-gated L-type Ca^2+^ channels (CaV) open as a consequence of membrane depolarization and are possibly involved in the abnormal Ca^2+^ entry into muscle cells from the extracellular space. The activity of CaV1.2 is increased in cardiomyocytes of 4–6-months-old mdx mice and determines enhanced Ca^2+^ entry, leading to abnormalities in cardiac electrophysiology and arrhythmia in DMD [64].

The abnormal Ca^2+^ uptake in the muscle fiber is also correlated with dysfunctions of store-operated calcium channels (SOCCs). SOCCs are sarcolemmal Ca^2+^ channels whose activation is stimulated by the depletion of intracellular Ca^2+^ depots, including primarily the SR, the main depot of Ca^2+^ ions in the muscles, and mitochondria. SOCCs open in response to reduced SR Ca^2+^ concentrations and their purpose is restoring intracellular calcium deposits [64]. In dystrophic muscles, Ca^2+^ release from the SR is increased and the uptake of Ca^2+^ ions by the SR is impaired [66]. Furthermore, a disruption of mitochondrial ion channels promotes the release of Ca^2+^ from mitochondria. These alterations contribute to Ca^2+^ overload in DMD and cause the chronic activation of SOCCs [64].

Two main proteins are responsible for SOCE: stromal interaction molecule (STIM1) and Orai1. STIM1 is located in the endoplasmic reticulum (ER) membrane and acts as a Ca^2+^ sensor, sensing the depletion of intracellular Ca^2+^ depots and subsequently activating Orai1 via direct physical interaction. Orai1 is a plasma membrane Ca^2+^ channel that allows the influx of Ca^2+^ ions in muscle cells [67]. In dystrophic muscle cells, the activity and expression of STIM1 and Orai1 is enhanced, contributing to muscle Ca^2+^ overload in DMD [64]. SOCC inhibitors targeting STIM1-Orai1 effectively prevent Ca^2+^ overload and ameliorate the decline in the contractile performance in myotubes from DMD-patient-derived induced pluripotent stem cells [68]. In addition, crossing mdx mice with a muscle-specific Orai1 knockout mice (mdx-Orai1 KO mice) normalizes Ca^2+^ homeostasis and promotes sarcolemmal integrity/stability, improving dystrophic muscle pathology [69].

Other dysfunctional channels that contribute to an excessive calcium entry into dystrophic muscle fibers are transient receptor potential channels (TRPCs). TRPCs located on the sarcolemma are activated by either the depletion of calcium stores or the stretching of the membrane. The expressions of TRPC1, TRPC3, and TRPC6 are increased in the skeletal, cardiac, and smooth muscles of mdx mice [64]. TRPC1 interacts with DAPC and, according to a model for TRPC1 regulation, the DAPC acts as a scaffold for the signaling molecules that regulate channels composed by TRPC1 and various TRPC isoforms [66]. In DMD, the loss of DAPC integrity determines the loss of this regulation and enhances the sensitivity of these channels to mechanical activation, leading to an increased Ca^2+^ entry [70]. In mdx mice, TRPC1 expression levels correlated with the severity of muscle phenotype [66]. Millay et al. demonstrated that the overexpression of TRPC3 and the subsequential increased Ca^2+^ influx are sufficient to induce a dystrophic phenotype nearly identical to that observed in DGC-lacking dystrophic disease models in healthy wild-type mice. Furthermore, the transgene-mediated inhibition of TRPC channels in mice significantly decreased Ca^2+^ influx and the symptoms of dystrophic disease [71]. A study using DMD^mdx^ rats showed that Pyr10, a specific inhibitor of TRPC3, reduced Ca^2+^ sarcolemmal permeability and mitigated the dystrophic phenotype [72]. In mdx mice and DMD patients, TRPC6 levels are increased both in the cardiac and skeletal muscles [64]. A recent study demonstrated that the gene deletion or the pharmacological inhibition of TRPC6 ameliorated skeletal and cardiac muscle defects and bone deformities in double knockout mice for both dystrophin and utrophin (mdx/utrn−/−), constituting a severe DMD model [73]. In addition to canonical TRPCs, another channel belonging to the TRPC family, the transient receptor potential vanilloid type 2 (TRPV2), may contribute to increase the intracellular Ca^2+^ concentration in DMD. TRPV2 is normally localized in the membrane of organelles, but in dystrophin-deficient muscle, it translocates to the sarcolemma [64].

### 5.2. Sodium

In dystrophic muscle cells, alterations in ion levels extend beyond calcium, involving interactions between the DAPC and sarcolemmal voltage-gated sodium (NaV) channels (NaV1.4 and NaV1.5 in the skeletal and cardiac muscles, respectively). The dysfunction of these channels occurs in DMD due to impaired DAPC function. In the skeletal muscles of mdx mice, intracellular sodium (Na+) levels are elevated due to the enhanced conductive properties of NaV1.4 channels, despite a reduction in NaV1.4 protein levels. The normalization of Na+ influx and the restoration of muscle fiber integrity in mdx muscle was achieved by inhibiting NaV1.4 channel activity with tetrodotoxin. Conversely, the cardiomyocytes of mdx mice exhibited a reduced Na+ influx correlated with decreased NaV1.5 protein levels, contributing to altered cardiac electrophysiology and cardiomyopathy development. Additionally, the increased intracellular Na+ levels in dystrophic muscles result from the up-regulated activity of Na+/H+ exchangers (NHE) type 1. The inhibition of NHE-1 with cariporide and 5-(N-ethyl-N-isopropyl)-amiloride reduced muscle cell degeneration in mdx mice. The resultant intracellular Na+ overload indirectly contributes to further increases in intracellular Ca^2+^ levels [64]. In healthy muscle, the Na+/Ca^2+^ exchanger (NCX) pumps the excess Ca^2+^ out of the cell, transporting Na+ into the cell. However, in cases of cytosolic Na+ overload, as observed in dystrophic muscles, NCX operates in reverse mode, expelling excess Na+ from the cell while pumping Ca^2+^ into the myoplasm. Furthermore, in dystrophic muscles, an enhanced Ca^2+^ release from the SR can induce NCX to function in reverse mode, exacerbating Ca^2+^ overload [66].

### 5.3. Potassium

In the cardiomyocytes of mdx mice, the outward K+ currents remain unchanged, but the inward rectifier K+ currents, mainly mediated by Kir2.1 channels, are reduced. In the cardiomyocytes of dystrophic dogs, a reduction in transient outward K+ currents was detected. In addition, the KATP subunit Kir6.2 is associated with dystrophin, and this association is lost in DMD, leading to a relevant reduction in KATP channel activity and in KATP currents in cardiomyocytes. K+ channels in the heart are fundamental to repolarize the cardiomyocytes and synchronize the electrical and contractile mechanisms, maintaining the rhythm of contractions. Indeed, in DMD, the alteration of K+ homeostasis contributes to the development of arrhythmia and various forms of cardiomyopathy [64].

## 6. Sarcoplasmatic Reticulum (SR)

In addition to dysfunctional sarcolemmal calcium channels, the dysregulation of the ion channels of intracellular organelles contributes to the development of DMD. The SR is one of the main intracellular Ca^2+^ depots, and its Ca^2+^ concentrations reach ∼0.4–0.5 mM [66]. During muscle contraction, the activation of RyR allows Ca^2+^ release from the SR. Ca^2+^ is subsequently pumped back to the SR by SERCA during relaxation. The release of Ca^2+^ from the SR is impaired in both cardiac and skeletal muscles in DMD, since RyR has been shown to become leaky. In dystrophic skeletal muscles, RyR1 leakage is due to post-translational modifications, in particular, a progressive S-nitrosylation of RyR1, combined with a depletion of *calstabin 1*, a critical regulator that maintains the closing state of RyR. These events are responsible for RyR1 Ca^2+^ leaks. Similarly, the S-nitrosylation of RyR2 and the depletion of *calstabin 2* determines Ca^2+^ leaks from the SR in the dystrophic cardiac muscle, likely triggering cardiac arrhythmias in mdx mice. The RyR stabilizer S107 inhibited the SR Ca^2+^ leak and prevented fatal sudden cardiac arrhythmias in vivo. Furthermore, the phosphorylation and oxidation of RyR2 could contribute to SR Ca^2+^ leaks via RyR2 in the mdx heart [66].

SERCA activity is significantly reduced in dystrophic skeletal muscles and cardiomyocytes. The repression of SERCA functioning is greater in mdx/utrn−/− mice and DBA/2J mdx than in the classical mdx mouse, contributing to a more prominent Ca^2+^ overload and a more severe phenotype [64]. A recent study showed that a single systemic delivery of SERCA2a with adeno-associated virus in 3-months-old mdx mice enhanced motility test performance and normalized electrocardiograms [74]. The transgenic induction of SERCA1 overexpression in the skeletal muscles of mdx and mdx:utr−/− mice ameliorated the DMD phenotype [64].

The ability of SERCA to uptake Ca^2+^ is significantly influenced by the expression of small-molecular-weight membrane proteins that modulate its function. These SERCA regulators are distinct in negative—including phospholamban (PLN), sarcolipin (SLN), and myoregulin (MLN)—and positive, such as dwarf open reading frame (DWORF), ones. Recently, a role of small ubiquitin-like modifier type 1 (SUMO-1) in defective cardiac SERCA activity in heart failure emerged [66]. It is worth mentioning that PLN levels remain unchanged in dystrophic heart and skeletal muscles: interestingly, crossing mdx mice with PLN knockout mice exacerbates mdx cardiomyopathy [75]. SLN is significantly up-regulated in the skeletal muscles and the ventricles of DMD patients and animal models. The partial or complete elimination of SLN ameliorates dystrophic pathology in a mdx/utrn−/− mouse model [76]. The expressions of MLN, DWORF, and SUMO-1 in dystrophic cardiac and skeletal muscles are yet to be analyzed in detail [66].

Mitochondria play a key role in the maintenance of Ca^2+^ homeostasis in striated muscles. The outer membrane of mitochondria allows the free diffusion of ions, while the transport of most compounds is mediated by voltage-dependent anion channels (VDACs). The inner mitochondria membrane presents various ion channels selectively permeable for specific ions and metabolites [64]. Mitochondria capture Ca^2+^ mainly via the mitochondrial Ca^2+^ uniporter (MCU) and extrude Ca^2+^ through the mitochondrial Na+-Ca2+-Li+ exchanger (mtNCLX). In response to elevated cytoplasmic Ca^2+^ concentrations, the uptake of Ca^2+^ in mitochondria is increased as a compensatory mechanism aimed at normalizing cytoplasmic Ca^2+^ levels [66]. However, a recent study highlighted a significantly reduced intensity of the Ca^2+^ uniport in the skeletal muscle mitochondria of mdx mice, apparently contributing to an increase in myoplasmic Ca^2+^ levels. Conversely, an enhancement in the Ca^2+^ uniport efficiency was detected in the heart mitochondria of mdx mice. Dystrophic skeletal and cardiac muscles are both affected by severe dysfunctions in the calcium-buffering capacity of mitochondria [64].

## 7. The Involvement of the Gut Microbiota in DMD

Gastrointestinal manifestations are common in DMD patients, including altered motility, constipation, pseudo-obstruction, and acute dilatation: they are mainly ascribed to the atrophy of smooth muscle layers [77] and can determine insufficient fluid and caloric intake, leading to gut microbiota dysbiosis [5]. Additionally, gut bacterial community alterations could be exacerbated by DMD-associated conditions that further promote intestinal dysbiosis. For instance, the sedentary lifestyle and the use of antibiotics prescribed to DMD patients to treat respiratory infections have deleterious consequences on the gut microbial status [78]. Intriguingly, the dysfunctions observed in the dystrophic intestine support the previously mentioned possibility that bacterial metabolites (or even bacteria themselves) translocate from the gut into the circulation. The absence of dystrophin significantly alters the gastrointestinal surface, resulting not only in disrupted ion absorption but also in the loss of protective cells, such as enterocytes and macrophages, and the formation of membrane tears. These conditions could facilitate bacterial passage and the subsequent activation of cytotoxic B and T lymphocytes and Tregs, along with the detrimental secretion of pro-inflammatory signals. Alternatively, they enhance the development of fibrosis in the smooth muscle of the intestine, impair intestinal contractility, reduced fecal excretion, and delay the transit time of the fecal material [79] (Figure 1).

To date, there is no cure for DMD, but therapies able to delay the onset of symptoms or slow down the progression of the pathology have been developed in recent decades [62]. Currently, glucocorticoids (GCs) like prednisone are the standard treatment, although they come with serious side effects affecting various systems [80]. GC consumptions also modifies the gut microbiota composition, contributing to the rise in side effects: the treatment with prednisone can cause a shift in phyla, favoring both pro-inflammatory bacteria and bacteria that produce B-cell superantigens and modulate T-cell differentiation [5].

Considering the role of nutrition-based approaches in modulating chronic inflammatory responses by influencing the gut microbiota, nutraceutical supplementation in mdx mice demonstrated a scavenger activity and reduced ROS production. Our findings also revealed a mobilization of bone marrow-derived CD45+ stem cells expressing the surface markers CD34, which regulates the attachment of stem cells to the bone marrow ECM, and stem cell antigen-1 (Sca-1), considered the most reliable marker for identifying hematopoietic stem cells [81]. Supplementation with a branched-chain amino acid-enriched mixture (BCAAem) and a mixture of flavonoids and docosahexaenoic acid showed positive effects on muscle phenotype and function [82,83]. Clinical studies in dystrophic individuals revealed that the oral administration of natural polyphenols improved performance and reduced inflammatory markers [84]. Interestingly, factors such as a sedentary lifestyle and antibiotic use further exacerbate gut microbiota dysbiosis in DMD [78].

Alterations in the colonic mucosa and changes in the richness of gut microbiota were noted in mdx mice, particularly characterized by a reduced diversity and the dominance of a select few bacterial species, including *Parasutterella*, *Alistipes*, *Rikenella*, and *Prevotella*. Notably, *Prevotella* has previously been linked to metabolic and inflammatory changes in the gut [85]. Accordingly, we demonstrated the effects of microbiota modulation on skeletal muscle morphology: in particular, we assessed the shift in fiber type toward an oxidative phenotype, the alterations of metabolic pathways dependent on AMPK, and a significant down-regulation of inflammation and fibrosis [79]. Indeed, we performed the gnotobiotic colonization of mdx mice with C57Bl microbiota, leading to a marked limitation of immune responses, improvement in muscular metabolism, and significant amelioration of skeletal muscle morphology and tetanic force. All things considered, the modulation of intestinal microorganisms could be evaluated as a feasible tool to both improve DMD patients’ symptoms and diminish muscular inflammatory cues [79].

Another recent study followed the microbial diversity, the plasma biomarkers, and the intestinal and muscular features of mdx mice through one year of age to better determine the mediators of skeletal muscle–intestine crosstalk [78]. Regarding the composition of the gut microbiota, they found that the abundances of *Bacteroidetes* and *Firmicutes* were similar throughout the age between mdx mice and the age-matched controls, while there was a significant up-regulation of *Actinobacteria, Proteobacteria*, and *Tenericutes* in dystrophic mice. In the plasma of mdx mice, they revealed an up-regulation of the common pro-inflammatory cytokines IL-6, TNF-α, and monocyte chemotactic protein-1 (MCP-1), while, on contrary, they assessed a dramatic decrease in adipokines [78]. As expected, the expression of different genes affecting the permeability of the intestine *Tight Junction Protein 1* and *2* (*TJP-1* and *TJP-2)* or those regulating gut bacterial metabolites—*branched chain amino acid transaminase 2 (Bcat2)* and *Adiponectin receptor 1 (AdipoR1)*—and inflammation—*tlr4*, *myd88*, and *Angiopoietin-like 4 (Angptl4)*—in skeletal muscles were altered in mdx mice, often increasing with age. This evidence clearly confirms that an increased intestinal permeability allows the transit of bacterial component toward the circulation, increasing the inflammatory conditions of mdx mice [78]. These findings enhance our comprehension of the intricate interplay among the gut microbiota, inflammation, and DMD pathology and present the gut microbiota as a fundamental metabolic regulator, highlighting potential therapeutic targets in the gut–muscle crosstalk for this challenging condition (Figure 1).

The precise regulation of the main ion currents is essential for proper muscle contraction and relaxation cycles. In dystrophin-deficient muscle cells, an excessive calcium ion intake results in calcium overload, impairing muscle fiber relaxation. Consequently, individuals with DMD and animal models often experience hypercontraction and deficits in muscle relaxation. The alteration of Ca^2+^ homeostasis in the myoplasm has also other consequences, including the impairment in muscle cell differentiation and the overactivation of Ca^2+^-dependent proteases (calpains) and phospholipases, leading to a significative proteolytic damage to cellular proteins and to the disruption of cell and organelle membranes [64]. These degradative pathways contribute to muscle cell death. Furthermore, an enhanced Ca^2+^ concentration induces mitochondrial dysfunction, which results in metabolic defects and mitochondrial-dependent necrosis [65]. Consequently, inflammation develops and contributes to promote the replacement of muscle tissue with connective and adipose tissues [64]. In response to intracellular ions’ overload, intracellular organelles, especially mitochondria and the SR, can undergo dysfunction or adaptive responses. Since these organelles play a key role in regulating ionic homeostasis, their alterations lead to the further dysregulation of ionic equilibrium, as well as metabolic dysfunction and the activation of cell death pathways [64]. Intriguingly, studies have shown that calcium supplementation in mice significantly alters the composition of the gut microbiota, particularly enriching species such as *Akkermansia, Acitenobacter*, and *Klebsiella* and influencing SCFA production [86].

## 8. Microbiota Interactors and Microbiota-Derived Metabolites Modulate the DMD Phenotype

### 8.1. PPARs Inhibition

Since PPAR-β/δ modulate the proliferation of slow oxidative muscle fibers, it was investigated whether the sustained expression of these PPARs in mdx mice ameliorated their pathological phenotype. In treated mdx mice, there was described the up-regulation of utrophin-A—that is structurally similar to dystrophin and can exert similar functions—as well as other components of DAG, α1-syntrophin, and β-dystroglycan [27]. Similarly, in mdx mice, the same PPARδ agonist in association with other drug-blocking AMPK activity improved muscle functions and diminished the expression of CKs and other proteins associated with muscular oxidative metabolism [87].

### 8.2. The Endocannabinoid System

The endocannabinoid system (ECS) is a fundamental modulator of neuronal transmission that exerts its functions on neurons, glia, immune cells, and blood–brain barrier cells; it is formed by endogenous cannabinoids (endocannabinoids) and their receptors and ligands. ECS is considered as a potential target for neurological diseases, as it mediates inflammation in Alzheimer’s and Parkinson’s diseases as well as amyotrophic lateral sclerosis [88]. In the skeletal muscle, one of the component of the ECS, the CB1 receptor, affects insulin sensitivity and substrate oxidation [89] while, more generally, the ECS controls metabolic function and immune responses [90] and energy homeostasis [91].

Manca et al. observed age-dependent modifications in intestinal endocannabinoidome (eCBome) gene expression in GF mice lacking a gut microbiota. A fecal microbiota transplant (FMT) from conventionally raised donor mice partially restored these changes in age-matched GF male mice, highlighting the direct impact of the gut microbiota on the host eCBome [92]. Consequently, the intestinal flora could influence the host organism through the modulation of endocannabinoid signaling. Accumulating evidence supports the involvement of endocannabinoids and their receptors in maintaining the gut barrier function, controlling gut permeability, regulating inflammation, and modulating autophagy [93]. The eCBome regulates tight junction proteins crucial for IEB function: its control over the gut barrier function operates mainly through a CB1-dependent mechanism [94], as CB1 antagonists act as gatekeepers, with 2-AG and 2-OG considered as such. Notably, 2-OG can bind to the G-protein-coupled receptor 119 (GPR119), expressed by enteroendocrine L cells in the intestine, which secrete glucagon-like peptide 2 (GLP-2), a hormone involved in maintaining the gut barrier function [93]. The gut microbiota, by modulating the ECS, partially regulates the preservation of IEB permeability and integrity, as SCFAs can enhance the intestinal levels of 2-AG and 2-OG [17].

SCFAs are intricately linked to inflammation and autophagy, both playing critical roles in DMD progression. Recently, Kalkan et al. proposed that the ECS is implicated in DMD onset and progression, suggesting a direct communication between the gut microbiota and the eCBome [53]. In mdx mice, a dysbiotic microbiota leads to inadequate butyrate production and circulation: this alteration results in an excessive endocannabinoid system activity, exacerbating inflammation and impairing autophagy in the muscles. Similarly, sodium butyrate (NaB) administration reduces skeletal muscle inflammation and autophagy deficits, as seen in primary human myoblasts isolated from DMD donors. Importantly, the effects of NaB on the mdx pathological phenotype were similar to those exerted by the glucorticoid deflazacort, whose modulation of the mdx gut microbiota was additionally evidenced. These findings highlight the dysregulation in the gut microbiota–endocannabinoid system axis [53].

### 8.3. Choline Inhibition

The supplementation of mdx mice with choline has been shown to ameliorate various pathological features. Treated dystrophic mice exhibited a significant reduction in fibrotic areas in the diaphragm muscle, accompanied by a decrease in both CD68+ pro-inflammatory macrophages and the transcription of pro-fibrotic genes. Furthermore, an improvement in SERCA maximal activity was observed, suggesting choline supplementation as a feasible adjunctive treatment [95]. Additionally, the up-regulation of choline was observed in the brains of 6-months-old mdx mice, similar to what has been observed in boys with DMD [96]. This up-regulation may be linked to dysfunctions in oxygen consumption and glucose metabolism [97].

### 8.4. Polyamines’ Inhibition

Mutations in the *Laminin-α2* gene determine the LAMA2-deficient congenital muscular dystrophy (LAMA2-CMD), an irreversible degenerative muscle disease leading to muscle weakness, neuropathy, and hypotonia in the lower limbs. In the LAMA2-CMD animal model, a down-regulation of two polyamine genes, *S-Adenosylmethionine decarboxylase* (*Amd1*) and *Spermine oxidase* (*Smox*), was observed in the severely affected tibialis anterior muscle. This down-regulation was associated with the extensive development of fibrosis and the expression of TGF-β [98].

### 8.5. Taurine Inhibition

Administering taurine in the diet of mdx mice led to significant improvements in cardiac function, accompanied by a reduction in IL-6 expression [99]. Likewise, the prenatal supplementation of mdx mice with taurine enhanced muscle strength in the dystrophic offspring by influencing the proteomic expression of matrix metalloproteinase 2 (MMP2) [100]. Therefore, it is widely recognized that modulating taurine expression in the context of DMD is crucial for regulating skeletal muscle metabolism [101], opening the way for the therapeutic use of taurine in combination with glucocorticoids [102].

## 9. Conclusions

Thanks to the current standards of care, some DMD patients live up to their fourth decade of life. In the absence of a curative treatment, prioritizing the prevention of the secondary processes that worsen DMD progression is crucial for enhancing the patients’ quality of life. We propose that, in DMD, intrinsic environmental factors (such as innate and adaptive immunity) and extrinsic factors (such as nutrition) are interconnected in a well-defined temporal and spatial manner, and their dysfunctions are the primary causes of chronic inflammatory conditions. The identified involvement of the gut–muscle axis in DMD pathophysiology opens avenues for innovative therapeutic strategies targeting specific intestinal microorganisms, microbial products, or metabolites to improve DMD symptoms. Cutting-edge methods are currently being employed to isolate and potentially target specific components of the gut microbiota that play a fundamental role in musculoskeletal health. The goal is to enhance their proliferation and the production of their metabolic products. Additionally, dietary interventions can serve as valuable co-adjuvants for DMD treatment, mitigating the effects of dystrophic progression by counteracting chronic inflammation. Although further studies are needed to better elucidate the role of pathways along with the gut–muscle axis in DMD development/progression, microbiota modulation could potentially offer therapeutic benefits similar to corticosteroids without adverse side effects. This approach will facilitate the development of precise interaction models to define the microbial–musculoskeletal system interactions that could potentially enhance the quality of life of DMD patients and reduce the costs associated with recurrent hospitalizations.

## Figures and Tables

**Figure 1 ijms-25-05589-f001:**
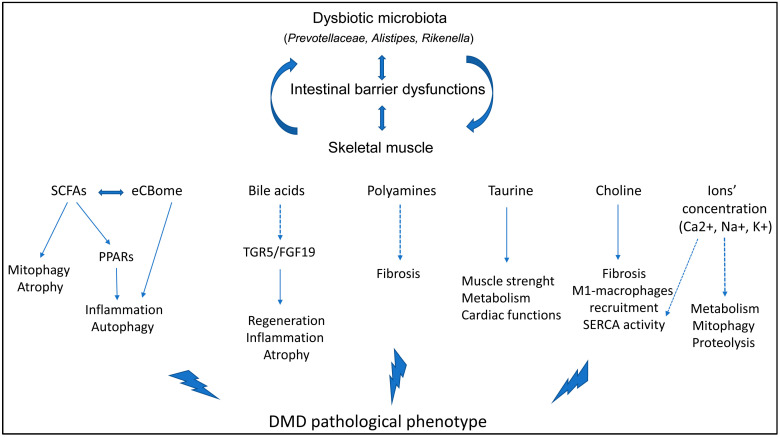
The combination of the reduction in gut microbiota richness and the up-regulation of certain strains, such as *Prevotellaceae, Rikenella*, and *Alistipes*, determines the dysbiosis in dystrophic mice. Taken together with a dysfunctional intestinal barrier, it leads to the uncontrolled passage of several microbiota-dependent proteins toward the skeletal muscle. This situation contributes to the proliferation of inflammatory cells (such as M1 macrophages) that, in turn, determines maladaptive autophagy and muscle regeneration and alterations in metabolism. The synergistic relationship among these events causes the development of skeletal muscle atrophy and fibrosis and the consequent wasting of muscle, worsening the DMD pathological phenotype. Dashed lines represent signaling pathways that are described in muscular diseases, but their microbiota-dependent effects on the mdx pathology have to be demonstrated yet. Abbreviations: eCBome: endocannabinoidome; PPARs: peroxisome proliferator-activated receptors; SERCA: sarco-endoplasmic reticulum calcium ATPase; TGR5/GP-BAR1: G protein-coupled bile acid receptor; FGF19: fibroblast growth factor 19.

**Table 1 ijms-25-05589-t001:** The immunomodulatory effects of gut microbiota-derived proteins on the pathological features of dystrophic skeletal muscle.

Protein Intermediates or Metabolites/Sources	Role	Effects on the Skeletal Muscle	References
Peroxisome proliferator-activated receptors (PPARs)/Gene expression	α: cellular uptake, energy homeostasis, and inflammation.	Expression of adipogenic genes, fatty acid metabolism, atrophy, inflammation, and myofiber type switching.	[24,25,26]
	β/δ: energy expenditure, tissue regeneration and repair, and inflammation.		
	γ: energy homeostasis, adipogenesis, triglyceride storage, and deposition of fat.		
Short-chain fatty acids (SCFAs)/Bacterial fermentation of non-digestible carbohydrates	Anti-inflammatory effects through GPR43 binding and the modulation of HDAC activity and cytokine, and PGE2 synthesis.	Regulation of lipid and carbohydrate expression, protein metabolism, and blood flow; and anti-inflammatory properties.	[18,20,31,34]
Choline (and its derivatives)/Dietary sources	Liver and neural metabolisms, and metabolism of membrane constituents.	Integrity of skeletal muscle cells and synthesis of acetylcholine (ACh).	[36,37]
Polyamines/Decarboxylation of amino acids	Cell growth, metabolism, and development; and antioxidant, anti-inflammatory, and anti-apoptotic effects.	Regulation of atrophy and muscle fiber size via mTORC1.	[39,40,41]
Tryptophan (Trp) and its metabolites/Dietary sources and gut microbiota	Inflammation in the gastrointestinal tract, nervous system, and muscles.	Age-related frailty and sarcopenia, and atrophy.	[42,43]
Bile acids (BAs)/Cholesterol	Digestion, absorption of dietary lipids and fat-soluble vitamins; lipid and glucose metabolisms; and systemic inflammation.	Regulation of skeletal muscle mass and function; and the involvement in atrophy and sarcopenia.	[6,44,46,47,48,49,51]
Taurine (2-aminoethanesulfonic acid)/Dietary sources	Osmotic pressure of different tissues and oxidative stress; and cytoprotective and anti-aging activities.	Myofiber necrosis, protein folding, and mitochondrial activity.	[52]

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
