# Peer review of "Exploring the Gut Microbiota–Muscle Axis in Duchenne Muscular Dystrophy"

_ijms, 2024, doi:10.3390/ijms25115589_

Round 1

Reviewer 1 Report (Previous Reviewer 1)

Comments and Suggestions for Authors

Now the manuscript has been significantly improved with the new information. 

Comments on the Quality of English Language

Please go over the manuscript to correct any minor linguistic errors.

Author Response

As suggested by the Referee, we carefully revised the Manuscript for typos and linguistic errors

Reviewer 2 Report (Previous Reviewer 2)

Comments and Suggestions for Authors

Dear authors,

You manuscript has been significantly improved and I recommend it for publication.

One small thing that I think you should change: it sounds strange when you discuss a positive effect on inflammation on lines 338-339. I guess you mean anti-inflammatory effects of mentioned bacteria. Please, rephrase.

Kind regards

Author Response

As suggested by the Referee, we modified the sentence in the new version of the Manuscript

Reviewer 3 Report (Previous Reviewer 3)

Comments and Suggestions for Authors

The authors have significantly expanded the manuscript and improved its presentation.

Author Response

We thank the Referee for the comments regarding the revised version of our Manuscript

This manuscript is a resubmission of an earlier submission. The following is a list of the peer review reports and author responses from that submission.

Round 1

Reviewer 1 Report

Comments and Suggestions for Authors

No comments.

Reviewer 2 Report

Comments and Suggestions for Authors

Reviewer comments:

Thank you for the opportunity to review the manuscript entitled: ‘Exploring the gut microbiota-muscle axis in Duchenne Muscular Dystrophy. It is a review paper, focusing on gut-muscle axis. The paper is elucidating the existence of a gut-muscle axis, linking together available information regarding alterations in gut microbiota composition, intestinal epithelial permeability and inflammation to progression of muscular dystrophy, in particular, Duchenne Muscular Dystrophy (DMD).

I found the area of gut-muscle axis is very exciting and novel, which is giving opportunity for the readers to learn more about this axis.

Major comments:

1. The manuscript is based on 63 published papers including both, original articles and review papers as well. I highly recommend to create a table, showing in the columns what kind of model has been described (e.g. human study, animal models, cells models etc), the key-findings (e.g., alteration in which bacteria, immune signals, etc) and the references and type of paper (e.g. original study or reviews).

2. Authors are repeating several times word inflammatory immunity, however it would be much better to specify what they are meaning, e.g. inflammatory signals or activation of adaptive immune system with recruitment of T-cells? For instance, on line 316, specify which type of cell are recruited, instead of just saying ‘ inflammatory cells’ .

Line 101: ‘activation of immune system’ – please explain what do you mean.

In additional, authors need to specify type of inflammation they are referring to, - systemic or local.

3. Authors mentioned several times possible translocation of gut bacteria into the blood circulation, reaching the muscle tissue (for instance already in the Abstract). Do you have references for that? If not, please keep discussion about passage of microbial products only (e.g. LPS etc).

4. Figure 1: I disagree with the wording – ‘ Decay of intestinal permeability’. I think, the authors meaning decline in intestinal barrier function, which is in turn increase permeability (=intestinal leakage). Please, correct or rephrase.

5. Please, be more specific in referring of bacterial species or genera. For instance, on line 324, … species, such as Prevotella, …. Specify these species and include the reference for the whole sentence.

6. Explain mdx abbreviation in the text when first mentioned.

7. Language needs to be improved.

Minor comments:

Line 247. What authors mean by ‘ positive effects on oxidative stress…’ Do you mean, ‘positive effects on health by counteracting effects on …? Please, rephrase.

Line250, please specify strains of Lactobacillus and Bifidobacterium, which have positive effect on muscles.

Line 343. Abbreviations should be used after the full name description (e.g. tight junction protein 1 and 2, TJP-1, TJP-2 respectively).

Comments on the Quality of English Language

Some times it was difficult to understand what authors mean in the sentences. I recommend Moderate editing of English language.

Reviewer 3 Report

Comments and Suggestions for Authors

Mostosi et al provide a well-structured review summarizing the limited data available on the gut microbiota-muscle axis under physiological and pathological conditions. The authors focus on Duchenne muscular dystrophy. This hereditary pathology, initially caused by the loss of the dystrophin protein (primarily in muscle tissue), is also accompanied by a complex of disorders leading to the rapid progression of the pathology and early death of patients. One of the key phenomena is also dysbiosis and disruption of intestinal microflora. I have several recommendations to the authors:

1. Lines 265-269. It is also necessary to note the important regulatory role of dystrophin. This protein also regulates the activity of multiple sarcolemmal ion channels (calcium, sodium, potassium) and its loss leads to a significant change in their function and disruption of ion homeostasis. According to recent data, this mechanism plays a more significant role in the development of DMD than mechanical ruptures of the sarcolemma, which can spontaneously close. Recent reviews highlight this problem well.

2. Line 281. Stress and dysfunction of the sarcoplasmic reticulum should also be noted.

3. Line 308. I recommend that the authors note the effects of another glucocorticoid deflazacort on the gut microbiota (it's shown in PMID: 36594243).   

4. I recommend that the authors provide a more informative illustration demonstrating differences in the gut microbiota-muscle axis in normal and DMD conditions. The authors should also reflect in more detail on changes in the composition of gut microbiota in different DMD models; this information is not focused in the main text.